# Hypothalamic deiodinase type-3 establishes the period of circannual interval timing in mammals

Calum Stewart[1], T Adam Liddle[1], Elisabetta Tolla[1], Jo Edward Lewis[2], Christopher Marshall[3], Neil P Evans[1], Peter J Morgan[4], Fran JP Ebling[5], Tyler J Stevenson[1,6]*

[1]School of Biodiversity, One Health, and Veterinary Medicine, University of Glasgow, Glasgow, United Kingdom; [2]Institute of Metabolic Science-Metabolic Research Laboratories & MRC-Metabolic Diseases Unit, University of Cambridge, Cambridge, United Kingdom; [3]School of Physiology, Pharmacology and Neuroscience, University of Bristol, Bristol, United Kingdom; [4]Rowett Institute, University of Aberdeen, Aberdeen, United Kingdom; [5]School of Life Sciences, University of Nottingham, Nottingham, United Kingdom; [6]University of Glasgow, Glasgow, United Kingdom

*For correspondence:
tyler.stevenson@glasgow.ac.uk

Competing interest: The authors declare that no competing interests exist.

## eLife Assessment

This study provides **important** findings on the understanding of circannual timing in mammals, for which iodothyronine deiodinases (DIOs) have been suggested to be of critical importance, yet functional genetic evidence has been missing. The authors **convincingly** implicate dio3, the major inactivator of the biologically active thyroid hormone T3, in circannual timing in Djungarian hamsters, using a combination of correlative and gene knock-out experiments; thus this provides key insights into the evolution and function of animal annual timing mechanisms.

**Abstract** Animals respond to environmental cues to time phenological events, but the intrinsic mechanism of circannual timing remains elusive. We used transcriptomic sequencing and frequent sampling of multiple hypothalamic nuclei in Djungarian hamsters to examine the neural and molecular architecture of circannual interval timing. Our study identified three distinct phases of transcript changes, with deiodinase type-3 (*Dio3*) expression activated during the early induction phase. Subsequent work demonstrated that targeted mutation of *Dio3* using CRISPR–Cas resulted in a shorter period for circannual interval timing. Hamsters that are non-responsive to short photoperiods and fail to show any winter adaptations do not display changes in *Dio3* expression and do not show any change in body mass or pelage. Our work demonstrates that changes in *Dio3* induction are essential for setting the period of circannual interval timing.

## Introduction

Phenology of key life history traits is common across plant (*Piao et al., 2019*) and animal kingdoms (*Cohen et al., 2018*). The annual changes in day length are the predominant environmental cue which animals use to time seasonal life history transitions (*Stewart et al., 2022*; *Pérez et al., 2019*; *Wood and Loudon, 2014*). Plants and animals also exhibit endogenous circannual timing in the absence of any change in environmental cues. For example, bird migration (*Gwinner and Dittami, 1990*), mammalian hibernation (*Pengelley and Fisher, 1957*), and reproduction (*Woodfill et al., 1994*;

*Lincoln et al., 2006*) are all driven by robust intrinsically generated *circannual clocks*, the cycle of which nearly matches a 12-month period. Some *circannual timers* estimate an interval period (e.g., 6 months) in which programmed changes in physiology and morphology occur in anticipation of the next season. Such interval timers are commonly observed as flowering in plants (*Duncan et al., 2015*), diapause in insects (*Denlinger, 1974*), and spring emergence in rodents (*Prendergast et al., 2004*). Interval timers are typically characterized by having light-dependent induction, maintenance, and recovery phases.

Annual mammalian life history transitions are typically associated with major changes in energy demand (*Ricklefs, 1991*). While the melanocortin system, including neuropeptide y (*Npy*), agouti-related peptide (*Agrp*), and melanocortin receptors is known to regulate short-term changes in energy homeostasis (*Yeo et al., 2021*), the mechanisms and anatomical structures implicated in long-term, circannual variation in energy rheostasis are not well characterized. Somatostatin (*Sst*) (*Marshall et al., 2024*; *Petri et al., 2016*; *Petri et al., 2014*), proopiomelanocortin (POMC) (*Bao et al., 2019*; *Mercer et al., 2000*), and VGF nerve growth factor (*Barrett et al., 2005*) are known to be strong correlates of seasonal variation in energetic state. Tanycyte somas are essential for the integration of environmental and physiological signals required for circannual interval timing. Tanycytes are localized in the ependymal layer of the third ventricle and are highly sensitive to nutrient state (*Bolborea et al., 2020*) and receive photoperiodic signaling derived from thyrotropes in the pars tuberalis (*Hanon et al., 2008*; *Wood et al., 2020*). Previous work has established deiodinase type-2 (*Dio2*) and type-3 (*Dio3*) expression is anatomically localized to tanycytes and coordinate triiodothyronine-dependent annual transitions in physiological state (*Bao et al., 2019*; *Ebling and Lewis, 2018*; *Hanon et al., 2008*; *Murphy et al., 2012*; *Petri et al., 2016*; *Wood et al., 2020*). Here, we delineate the molecular architecture of circannual interval timing by the hypothalamus and pituitary gland and test the conjecture that *Dio3*, in the Djungarian hamster (*Phodopus sungorus*), is upregulated during the induction of circannual interval timing for energy rheostasis and functions to establish the duration (or period) of the circannual interval timer.

## Results

Male Djungarian hamsters were either kept in a long photoperiod (LP) control condition or moved from LP to short photoperiod (SP) conditions (*Figure 1*). Djungarian hamsters remain in LP phenotype unless exposed to SP. Pelage color, torpor, and body mass were used to monitor the induction, maintenance, and recovery phases of the circannual timer (*Figure 1a, c*; *Videos 1 and 2*). A full white pelage color was observed between 12 and 16 weeks after exposure to SP and gradually reversed to the LP agouti color after 28 weeks in SP. Hamsters engaged in torpor between 12 and 20 weeks in SP (*Video 2*). Massive, programmed changes in energy rheostasis were observed with a 30% reduction in daily food intake and a 20% decrease in average body mass by 12 weeks SP (*Figure 1c, d*). Both food intake and body mass started to reverse to LP conditions after 20 weeks in SP. Epididymal adipose tissue mass and plasma insulin concentrations paralleled the change in body mass (*Figure 1d*, *Figure 1—figure supplement 1*). These reversals in physiological state are indicative of the recovery phase of the circannual interval timer. GLP-1 did not display any change in circulating concentrations (*Figure 1e*). Plasma glucose concentration jumped sharply after 16 weeks in SP (*Figure 1e*), concurrent with the development of torpor in these animals (*Video 2*). The observed change in body mass, food intake, adipose tissue, and plasma insulin versus the lack of change in GLP-1 highlights the physiological distinction between a programmed rheostatic mechanism characteristic of the circannual interval timing of energy stability in the former, while the latter are driven by short-term homeostatic mechanisms (*Stevenson, 2024*).

We then used Oxford Nanopore transcriptomic sequencing to characterize the molecular changes associated with phases of the circannual interval timer within multiple individual hypothalamic nuclei and the pituitary gland (*Figure 1f, g*; *Figure 1—figure supplements 2–6* and *Source data 1–5*). The development of refractoriness to melatonin, indicative of the recovery phase of the circannual timer, has been shown to develop independently in different hypothalamic regions (*Freeman and Zucker, 2001*). Molecular data was collected from the mediobasal hypothalamus (MBH; *Bolborea et al., 2020*), dorsomedial hypothalamus (DMH; *Ebling et al., 2008*), and the paraventricular nucleus (PVN; *Bittman et al., 1991*), along with the pituitary gland (*Majumdar et al., 2023*). This experimental approach provides a high-throughput and high-frequency sampling resolution to comprehensively

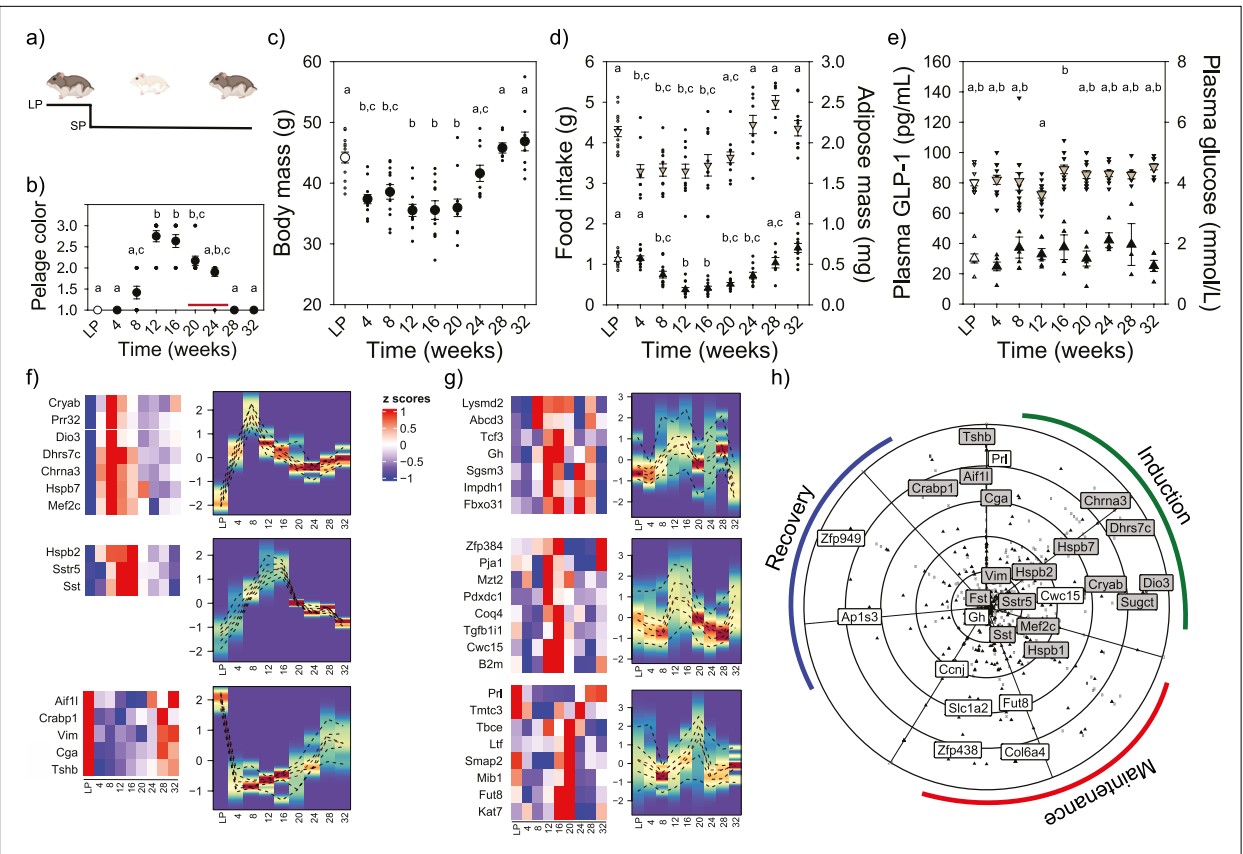

**Figure 1.** Molecular basis of circannual interval timing for morphology and physiology. Experimental design in which Djungarian hamsters were kept in long photoperiod (LP) or transferred to short photoperiod (SP) (**a**). SP-induced pelage color change from LP agouti to winter white after 12 weeks which reversed to agouti by 28 weeks ($F_{8,91}$ = 50.77; p < 0.001) (**b**). Torpor was identified in hamsters between 12 and 20 weeks SP exposure indicated by the red line (**b**). SP exposure induced significant reduction in body mass ($F_{8,91}$ = 13.428; p < 0.001) (**c**), food intake ($F_{8,91}$ = 10.860; p < 0.001; denoted as downward triangles) (**d**) and adipose mass ($F_{8,91}$ = 27.929; p < 0.001; denoted as upward triangles) (**d**). Plasma GLP-1 did not vary in response to SP manipulation denoted as upward triangles (**e**). SP exposure resulted in significant changes in plasma glucose around the onset of torpor; denoted as downward triangles ($F_{8,91}$ = 3.117; p < 0.05) (**e**). BioDare 2.0 heatmaps of mediobasal hypothalamus (**f**) and pituitary gland (**g**) transcripts from Djungarian hamster collected at 4-week SP intervals. Transcripts identified as highly rhythmic (FDR <0.1) showed three distinct phases of expression that coincide with the induction, maintenance, and recovery of circannual interval timing. Deiodinase type-3 (*Dio3*) was upregulated during the induction phase, whereas transcripts associated with energy stability (e.g., somatostatin [Sst and Sstr5]) were upregulated during the maintenance phase. All rhythmic transcripts reverted to the LP condition by 28 weeks SP exposure. Polar scatter chart of significant transcripts from mediobasal hypothalamus and pituitary gland provides a comprehensive seasonal clock for mammalian circannual interval timing across neuroendocrine tissues (**h**). The green line indicates the induction phase, the red line indicates the maintenance phase, and the blue line represents the recovery phase. Data presented in (**b-e**) are mean and standard error of the mean and evaluated using one-way ANOVA. Letters denote significant differences between treatment groups (p < 0.05) (**b–e**). In (**f, g**) the scale bar represents transcript expression as *z*-scores from 1 (upregulation in red) to –1 (downregulation in blue). Density heatmaps are adjacent to transcript expression heatmaps and display the average *z*-score expression of each individual cluster on the *y*-axis, and the graph shows percentile lines and density of *z*-score expression. Created with BioRender.com.

The online version of this article includes the following figure supplement(s) for figure 1:

**Figure supplement 1.** Plasma insulin reflects changes in transcript expression.

**Figure supplement 2.** Gene ontology analysis of mediobasal hypothalamus (MBH) sequencing reveals well-known seasonal pathways.

**Figure supplement 3.** Gene ontology analysis of pituitary gland unveils potential mechanisms for seasonal changes in protein processing and release.

**Figure supplement 4.** Paraventricular and dorsomedial hypothalamus sequencing unveils novel transcripts and widespread seasonal interval timing within the hypothalamus.

**Figure supplement 5.** Gene ontology analysis of paraventricular hypothalamic sequencing.

**Figure supplement 6.** Gene ontology analysis of dorsomedial hypothalamus suggests widespread immune involvement and cellular differentiation.

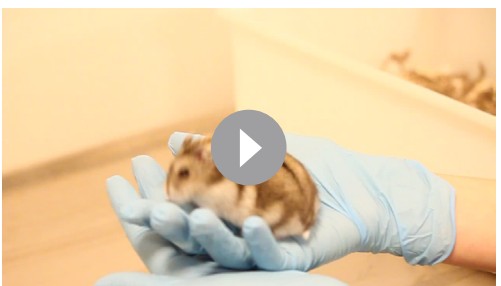

**Video 1.** Long photoperiod hamster – non-torpor state.
https://elifesciences.org/articles/106383/figures#video1

chart the induction, maintenance, and recovery of the mammalian circannual interval timer (*Figure 1h*). Biodare 2.0 identified 290 (*Source data 1*) transcripts in the MBH as endogenously rhythmic. Density heatmaps show an early and robust upregulation of transcripts by 8 weeks SP associated with the circannual interval induction (*Figure 1f*). Consistent with previous reports (*Petri et al., 2016*; *Bao et al., 2019*; *Yoshimura et al., 2003*), *Dio3* expression was significantly upregulated and clustered with the initial wave of transcript expression (*Figure 1f*, *Figure 1—figure supplement 1*). A second cluster of transcripts was upregulated during the maintenance phase occurring between 12 and 20 weeks in SP, which primarily consisted of somatostatin (*Sst*) and the cognate receptor subtype 5 (*Sstr5*). Increased *Sst* expression coincided with the reduction in body and adipose tissue mass reflecting the conserved role in inhibiting growth and metabolism (*Figure 1—figure supplements 1 and 2*). Finally, a third wave of transcripts became upregulated during the recovery phase and largely resembled the molecular landscape in LP hamsters (*Figure 1f*). qPCR assays for *Dio3* and *Sst* expression reflected the sequencing transcript count pattern providing independent replication of the sequencing analyses (*Figure 1—figure supplement 1*). In the pituitary gland, 250 transcripts were identified as rhythmic and included genes associated with secretagogue cells such as somatotrophs (*Gh*) and lactotrophs (*Prl*) (*Figure 1g*, *Figure 1—figure supplement 3*; *Source data 2*). A similar rhythmic expression was identified in other hypothalamic nuclei including the PVN (518 transcripts) (*Figure 1—figure supplements 4 and 5*; *Source data 3*), and the DMH (374 transcripts) (*Figure 1—figure supplements 4 and 6*; *Source data 4*).

## *Dio3*-dependent changes in triiodothyronine induced body mass loss via *Sst* expression

Treatment of Djungarian hamster with the somatostatin agonist pasireotide decreases body mass and is sufficient to drive changes similar to SP exposure (*Dumbell et al., 2017*; *Dumbell et al., 2015*). Further, certain SST receptor subtypes appear to be involved in driving seasonal torpor in Djungarian hamsters (*Scherbarth et al., 2015*). To determine if hypothalamic *Sst* expression reflects the programmed rheostatic change in energy state governed by circannual interval timing or homeostatic cues, tissues were collected from ad libitum fed hamsters under either LP control condition or the maintenance phase (12-week SP). To induce a negative energetic state, hamsters experienced an acute overnight food restriction (FR; 16 hr) (*Figure 2a*). As anticipated, body mass decreased approximately 30% after exposure to SP (*Figure 2b*, closed symbols); homeostatic negative energy state challenges by food restriction reduced body mass by 1.5 g on average for both LP (open symbols) and SP conditions (*Figure 2c*). Epididymal adipose tissue mass was higher in LP compared to SP but did not decrease in response to overnight FR (*Figure 2—figure supplement 1*). Plasma insulin was significantly reduced in response to SP treatment and FR, suggesting circulating levels are an output of both rheostatic and homeostatic signals (*Figure 2—figure supplement 1*). The two manipulations permit the ability to dissect the impact of rheostatic changes associated with circannual interval timing from those involved in short-term homeostatic changes in energy balance. Mediobasal hypothalamic *Sst* expression was significantly increased in SP conditions, but insensitive to homeostatic challenges in energy state (*Figure 2d*). In contrast, mediobasal hypothalamic *Npy* expression was significantly upregulated in food-restricted hamsters but was similar across LP and SP conditions (*Figure 2e*).

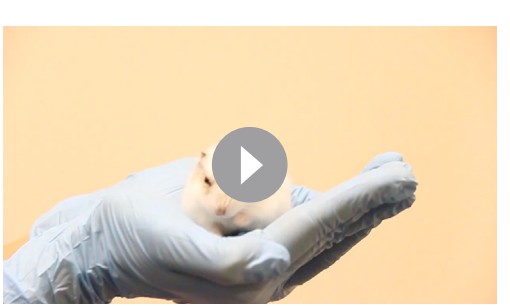

**Video 2.** Short photoperiod hamster – torpor state.
https://elifesciences.org/articles/106383/figures#video2

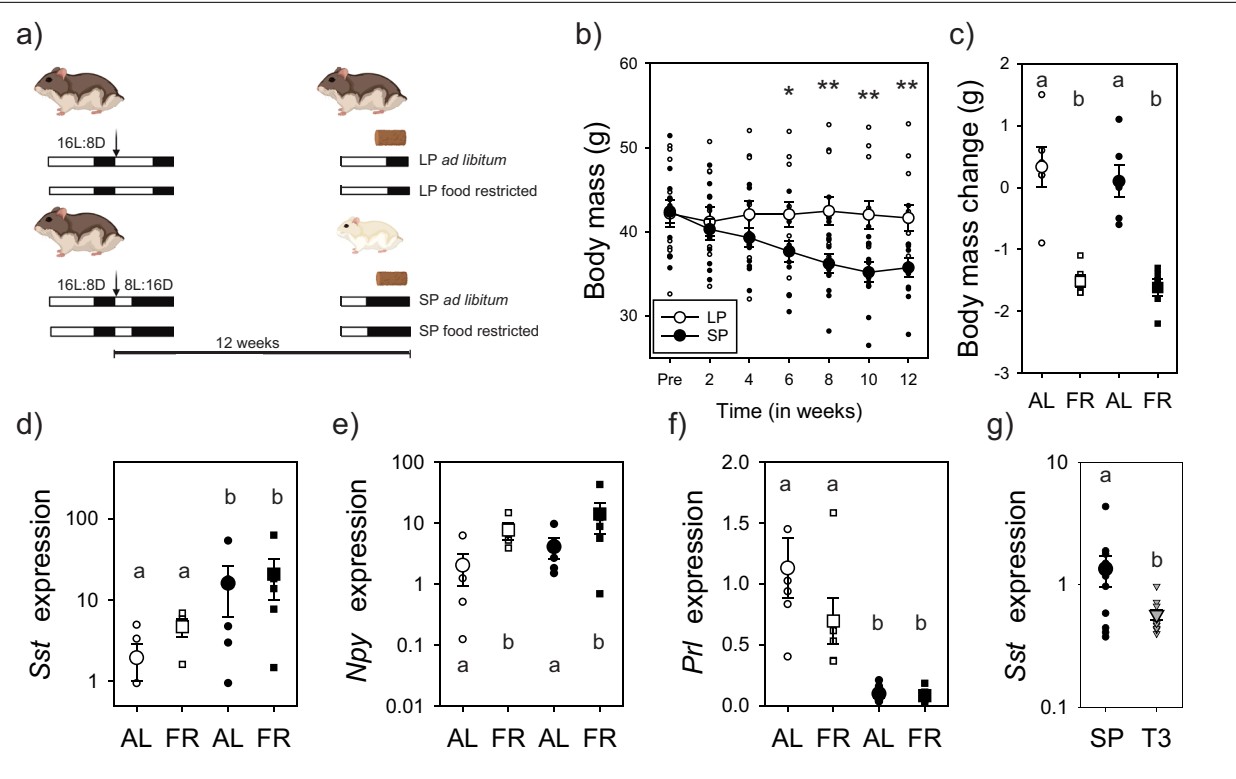

**Figure 2.** Somatostatin expression reflects programmed circannual interval timing that is dependent on *Dio3* regulation of local triiodothyronine signaling. The experimental design in which short photoperiod (SP) exposure induced rheostatic reduction in body mass after 12 weeks, and then hamsters experienced either an overnight food restriction (FR) for 16 hr or maintained food ad libitum (AL) (**a**). SP induced a significant reduction in body mass ($F_{1,22} = 3.85$; $p < 0.01$) (**b**). Food restriction further reduced body mass ($F_{1,22} = 25.46$; $p < 0.001$) (**c**). *Sst* expression in the mediobasal hypothalamus was significantly increased in SP ($F_{1,15} = 5.59$; $p < 0.05$) but did not change after manipulations in nutritional availability (**d**). *Npy* expression was increased after food restriction ($F_{1,15} = 6.12$; $p < 0.05$) but did not change with SP exposure (**e**). Prolactin (*Prl*) expression in the pituitary gland was downregulated in response to SP ($F_{1,20} = 28.53$; $p < 0.001$), and insensitive to food restriction. Twelve weeks of SP were found to increase *Sst* expression in the hamster mediobasal hypothalamus, and levels were significantly reduced in response to a single triiodothyronine (T3) injection (**g**). *p < 0.05 and **p < 0.01 denote significant difference between SP and long photoperiod (LP) conditions (**b**). Letters denote significant difference between treatment groups (p < 0.05) (**c–g**). Data presented in (**b-g**) are mean and standard error of the mean. Two-way repeated ANOVA was conducted on body mass values (**b**) and a two-way ANOVA was used to examine body mass change (**c**) and transcript expression (**d-g**). Created with BioRender.com.

The online version of this article includes the following figure supplement(s) for figure 2:

**Figure supplement 1.** Rheostatic mechanism controlling body mass change in Djungarian hamsters.

*Prl* expression was higher in LP compared to SP conditions (*Figure 2f*). *Gh* expression did not change in response to either SP-induced circannual changes in energy state or homeostatic manipulations (*Figure 2—figure supplement 1*). To establish whether mediobasal hypothalamic *Sst* expression is regulated by upstream *Dio3*-dependent changes in local thyroid hormone catabolism, SP housed hamsters received subcutaneous injections with either vehicle or triiodothyronine. A single triiodothyronine injection in SP hamsters was sufficient to reduce *Sst* expression compared to vehicle controls (*Figure 2g*). These data indicate SP-induced *Dio3* expression removes T3-dependent inhibition of *Sst* expression leading to long-term inhibition of body mass during the maintenance phase of circannual interval timing of rheostatic energy balance.

## *Dio3* dysfunction reduces the period of circannual interval timing

Next, we sought to establish the functional role of *Dio3* signaling for circannual interval timing of energy rheostasis in hamsters. Targeted genomic mutations of the *Dio3* gene localized to the MBH were achieved via intracerebroventricular (ICV) injection of CRISPR–Cas9 (*Dio3^{cc}*; *Figure 3—figure supplement 1*) to assess the functional role for circannual interval timing. Control hamsters were administered a blank Crispr–Cas9 construct (*Dio3^{wt}*). Two weeks following surgery, hamsters were

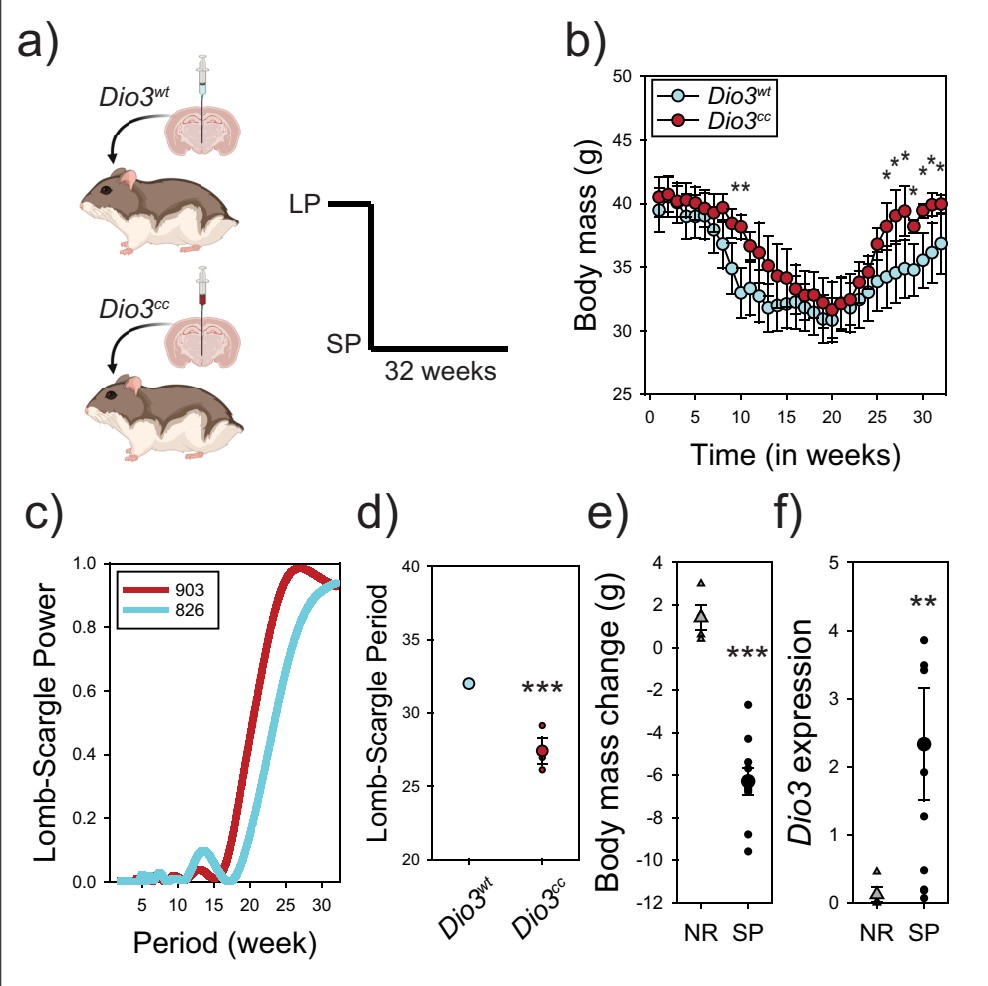

**Figure 3.** *Dio3* functions to time circannual interval duration in hamsters. Hamsters received intracerebroventricular injections to target *Dio3*-expressing cells in the tanycytes localized to the mediobasal hypothalamus (**a**). Crispr–Cas9 constructs were packaged into lentiviral vectors to generate blank control hamsters (*Dio3^wt*) or contain gRNAs that mutated the *Dio3* gene (*Dio3^cc*). Hamsters were then exposed to short photoperiod (SP) conditions and the circannual interval timer was assessed by monitoring body mass (**b**). *Dio3^cc* hamsters were slower to initiate interval timing as evidenced by higher body mass at 10 weeks SP exposure ($p < 0.001$) and recovered body mass quicker at 26–32 ($p < 0.05$) weeks. Lomb–Scargle analyses identified a significant reduction in the period of the circannual interval timer ($t_3 = 2.62$; $p < 0.05$) (**c, d**). Example *Dio3^cc* (hamster #903) and *Dio3^wt* (hamster #825) are presented in (**c**). A subpopulation of hamsters did not decrease body mass in response to SP exposure and was termed non-responsive (NR) ($t_{12} = 7.12$; $p < 0.001$) (**e**). *Dio3* expression in the mediobasal hypothalamus of NR hamster was nearly nondetectable after exposure to 8–12 weeks of SP exposure compared to SP control hamsters ($t_{12} = -3.78$; $p < 0.005$) (**f**). *p < 0.05, **p < 0.01, and ***p < 0.001. Data presented in (**b, d-f**) are mean and standard error of the mean. A two-way repeated ANOVA was conducted on body mass (**b**) and Welch's *t*-test used for (**d-f**). Created with BioRender.com.

The online version of this article includes the following figure supplement(s) for figure 3:

**Figure supplement 1.** Evidence of CRISPR modification in *Dio3^cc* and *Dio3^wt* Djungarian hamsters.

transferred to SP and programmed circannual changes in body mass (**Figure 3b**) and pelage were monitored (**Figure 3—figure supplement 1**). *Dio3^cc* hamsters had slower body mass loss and regained body mass significantly quicker during the recovery phase, compared to *Dio3^wt* controls. Similarly, the change in pelage color occurred later in *Dio3^cc* compared to *Dio3^wt*, and *Dio3^cc* was never observed to develop full white pelage (**Figure 3—figure supplement 1**). Lomb–Scargle period analysis of body mass identified that *Dio3^cc* hamsters have notably shorter duration (period = 26.959) compared to *Dio3^wt* (period = 32) (**Figure 3c, d**). There was no significant difference in terminal white adipose tissue

mass, or daily food intake. A subpopulation of hamsters was physiologically non-responsive (NR) to the SP manipulation and maintained the higher LP body mass (*Figure 3e*). *Dio3* expression in the MBH of these NR hamsters remained exceptionally low compared to responsive SP hamsters at 8 and 12 weeks following exposure to SP (*Figure 3f*). These data suggest that the inability to increase *Dio3* expression in NR hamsters acts as a form of naturally occurring dysfunction that prevents SP induction of circannual interval timing.

## Discussion

The molecular and cellular basis of circannual clocks and timers in vertebrates is not well characterized (*Helm and Stevenson, 2014*). Here, we sequenced the transcriptomes of three hypothalamic regions and the pituitary gland, key neuroendocrine structures which govern programmed circannual rheostatic and homeostatic processes. Our approach provided a high sampling frequency of the circannual interval timer in Djungarian hamsters. The findings produced a robust and comprehensive neural and molecular database that facilitates the delineation of a circannual interval timer in mammals. Several novel transcripts displayed a close correlation with the induction phase of the photoperiodic response. Among these was *Dio3*, conducive with the removal of thyroid hormone (e.g., thyroxine and triiodothyronine) being a key step in the transition to the winter phenotype. The photoinduced change in *Dio3* expression induces a long-term change in physiology and morphology, and our results demonstrate that it is insensitive to short-term alterations in response to other environmental cues (e.g., nutritional state). Functional manipulation of *Dio3* expression also delayed the induction and enhanced the reversal of the winter phenotype, suggesting that upregulation of *Dio3* acts to establish the period of circannual interval timing.

The transcriptomic data are consistent with evidence that shows *Sst* expression in the arcuate nucleus is one critical event for the maintenance of a reduced rheostatic energy state during the winter season (*Marshall et al., 2024*; *Petri et al., 2016*; *Petri et al., 2014*). In hamsters, *Sst* expression is upregulated after *Dio3* induction and is associated with the low body mass from weeks 12 to 20 before a downregulation in transcript levels coinciding with increased body mass. By increasing circulating triiodothyronine in hamsters maintained in SP conditions, our manipulation indicates that elevated hormone concentrations lead to the reduction in *Sst* expression in the MBH (*Figure 2*). The overall expression patterns provide the ability to develop a linear series of steps in which LP conditions have high local concentrations of triiodothyronine in the MBH (*Helfer and Stevenson, 2020*; *Murphy et al., 2012*), which reduces the levels of *Sst* expression resulting in consistently higher body mass. Exposure to SP induces *Dio3*, which catabolizes triiodothyronine leading to the removal of the inhibition of *Sst* expression and reduction in body mass. Local synthesis of triiodothyronine in the MBH by tanycytes is an evolutionary conserved process for timing photoperiod-induced transitions across the animal kingdom (*Ebling and Lewis, 2018*; *Lewis and Ebling, 2017*). Photoperiodic signals derived from the pars tuberalis via thyrotropin-stimulating hormone are essential to regulate *Dio3* (and *Dio2*) expression in tanycytes leading to increased triiodothyronine content in LP conditions for mammals (*Banks et al., 2016*, *Murphy et al., 2012*), birds (*Nakao et al., 2008*) and fish (*Lorgen et al., 2015*). The current limitation is understanding how SP signals regulate *Dio3* expression. Undoubtedly, the duration of melatonin secretion during the daily nocturnal phase is essential (*Helfer et al., 2019*). Future research is necessary to identify the conserved upstream signals that govern the timing of *Dio3* expression.

These data support the existence of a neuroendocrine pathway for the long-term rheostatic regulation, and another for short-term homeostatic control of energy stability (*Stevenson, 2024*). The rheostatic pathway consists of triiodothyronine (T3) signaling dependent on a *Dio3* switch to regulate *Sst* expression in the MBH, whereas short-term regulation of energy stability via nutrient availability is homeostatically regulated by well-established orexigenic (e.g., *Npy*) (*Yeo et al., 2021*). In response to the output of both pathways, peripheral effects may be mediated by acute and chronic changes in pancreatic insulin secretion (*Figure 4*). Overall, the findings provide a clear neural and cellular circuit for circannual interval timing of energy rheostasis and demonstrate *Dio3* as a gene critical for the control of seasonal life history transitions.

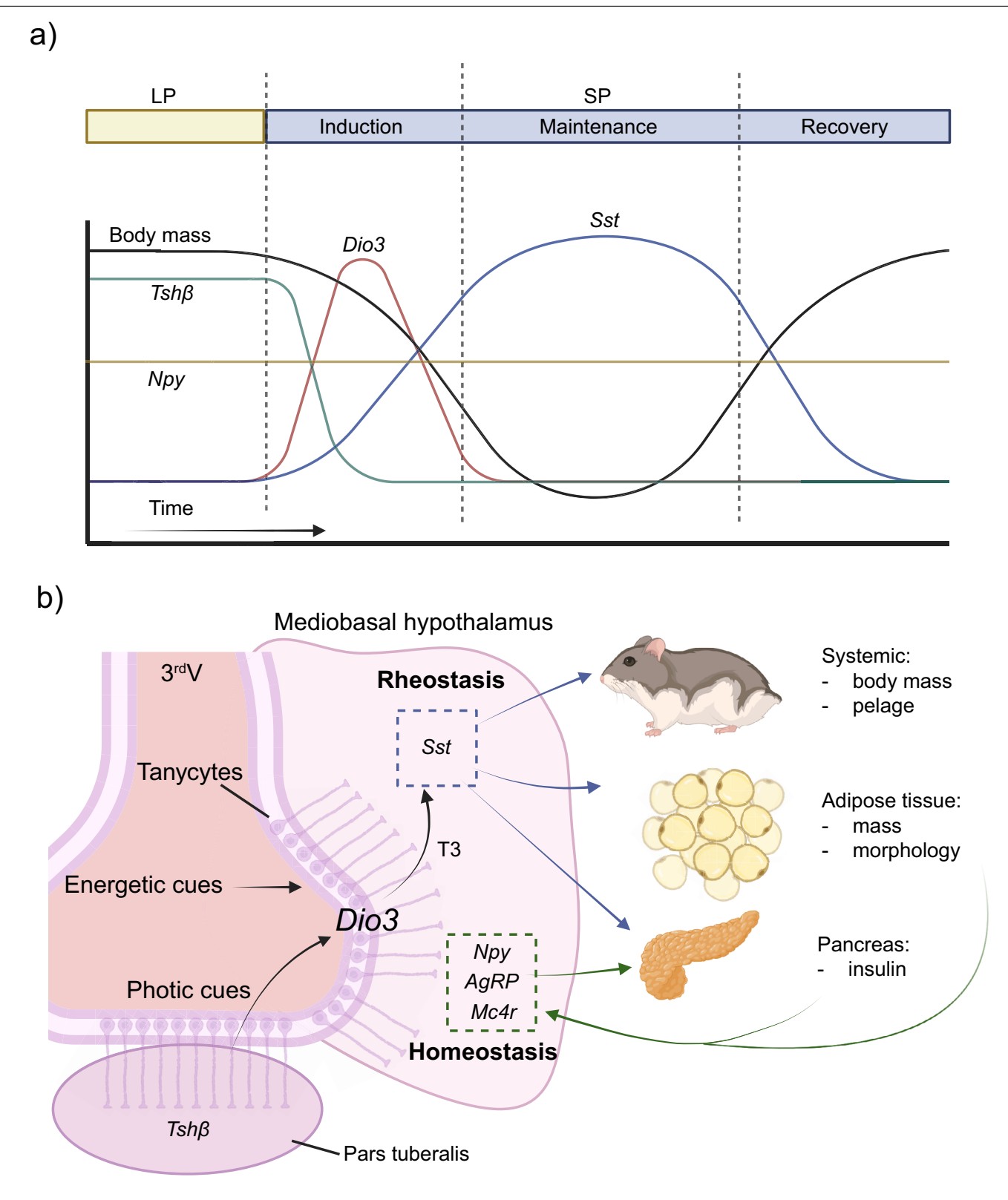

**Figure 4.** Schematic model for the mechanisms of circannual interval timing. (**a**) Hamsters held in long photoperiod (LP) have high body mass, thyrotropin-stimulating hormone beta (*Tshβ*), and low somatostatin (*Sst*) and deiodinase type-3 (*Dio3*). Transfer to short photoperiod (SP) results in the induction of circannual interval timing characterized by a reduction in body mass, low *Tshβ* in the pars tuberalis, and increased *Dio3* expression in tanycytes. The maintenance phase is associated with increased *Sst* expression, which serves to inhibit growth. The onset of the recovery phase is

*Figure 4 continued on next page*

*Figure 4 continued*

associated with a complete reversal in whole organismal physiology to the LP phenotype. (**b**) Tanycytes along the third ventricle in the mediobasal hypothalamus integrate photic cues derived from *Tshβ* and are sensitive to nutritional cues. The triiodothyronine (T3) output signal from tanycytes regulates *Sst* expression leading to long-term programmed rheostatic changes in body mass that serve to control circannual interval timing in multiple physiological systems. Conversely, homeostatic stability is maintained despite large-scale seasonal rhythms in body mass. Homeostatic energy balance is established by well-characterized circuits that include neuropeptide Y (*Npy*), agouti-related peptide (*Agrp*), and the melanocortin receptor 4 (*Mc4r*). Created with BioRender.com.

## Materials and methods

### Animals

All animals were obtained from a colony of Djungarian hamsters (3–8 months of age) kept at the University of Glasgow, Veterinary Research Facility. Animals were held at a room temperature of 21°C and a humidity of 50%. The colony room was kept on an LP (16L:8D). All hamsters had food (SDS [BK001]) and water ad libitum.

### Experiment 1 – Defining the circannual interval timer in hamsters

Adult male hamsters (N = 100) were required to assess the physiological and molecular basis of circannual interval timing in hamsters. A photoperiod reference group was kept on LP (n = 14) (8D:16L) conditions for the duration of the study. The SP treatment groups consisted of hamsters moved to SP (16D:8L) at 4 weeks intervals. The groups consisted of SP for 4 (n = 12), 8 (n = 12), 12 (n = 12), 16 (n = 11), 20 (n = 12), 24 (n = 10), 28 (n = 7), and 32 (n = 10) weeks. This photoperiod schedule reliably captures the induction, maintenance, and recovery of circannual interval timing in hamsters (*Prendergast et al., 2002*). Pelage score was determined using visual assessment and defined scale in which a score of 1 is the summer dark agouti coat; a score of 2 is a transition between agouti and white; and a score of 3 is a full winter white pelage (*Marshall et al., 2024*). Body mass was monitored to the nearest 0.1 g using an Ohaus portable scale (TA301). Food intake was measured by weighing the food pellets at the mid-point on two consecutive days to obtain a daily value. Food weight was measured using an Ohaus scale (TA301). Hamsters were killed between 2 and 4 hr after lights on by cervical dislocation followed by exsanguination. Adipose tissue was immediately dissected and measured using a Sartorius cp64 anatomical balance. A terminal blood sample was collected to measure glucose (Accu-Chek Performa nano blood glucose meter). Circulating levels of insulin and total glucagon-like peptide 1 (GLP1) using ELISA (MesoScale Discovery, UK) at Core Biochemical Assays laboratories, Cambridge, UK. Brain and pituitary gland tissue was quickly extracted and frozen on powdered dry ice and stored at –70°C. Epididymal white adipose tissue was dissected and weighed using Sartorius cp64 anatomical balance and stored at –70°C.

Brains were cut into 200 µm coronal sections using a Leica CM1520 cryostat. Anatomical structures (optic tract to the infundibular stem; Bregma –2.12 to –3.80 mm) were used to isolate the MBH, DMH, and the PVN. Bilateral tissue punches were performed using an Integra Miltex 1 mm disposable biopsy punch. Tissue punches were stored at –70°C until transcriptomic sequencing and confirmatory qPCR analyses (see below).

### Experiment 2 – Dissociating mechanisms that govern circannual rheostatic versus homeostatic energy stability

A total of N = 35 adult male hamsters were used to determine molecular signatures of rheostatic versus homeostatic energy stability. An LP control group was kept on LP for the duration of the study (n = 18). A subset of hamsters was moved to SP (n = 17) for 12 weeks to obtain animals in the maintenance phase of circannual interval timing. Food (SDS [BK001]) and tap water were provided ad libitum to both LP and SP animals for the 12 weeks duration. Body mass was measured every 2 weeks using a Traveler Ohaus portable balance. On the final night of the experiment, 50% of hamsters (i.e., LP = 9, SP = 9) were kept on food and water ad libitum. The other 50% of hamsters (i.e., LP = 9, SS = 8) served as the homeostatic treatment group and received an acute food restriction by removing all food. Overnight food restriction induces a robust negative energy state that reliably results in a 1- and 2-g loss in hamster body mass (*Bao et al., 2019*). Body mass was measured prior to food restriction and again before tissue collection on the subsequent day. Hamsters were killed between 2 and 4 hr after

lights on by cervical dislocation and then exsanguination. A terminal blood sample was collected to measure insulin (MesoScale Discovery, Core Biochemical Assays laboratories, Cambridge, UK). Body mass and epididymal white adipose tissue mass were measured using a Sartorius cp64 anatomical balance. Brains and pituitary glands were dissected and were immediately placed on dry ice and stored at –70°C. Brains were cut into 200 µM sections using a Leica CM1520 cryostat. Anatomical structures (optic tract to the infundibular stem; Bregma –2.12 to –3.80 mm) were used to isolate the MBH. Bilateral tissue punches were performed using an Integra Miltex 1 mm disposable biopsy punch. Brain tissue punches and pituitary gland samples were stored at –70°C until qPCR assay (see below).

## Experiment 3 – Sufficiency of triiodothyronine to induce LP neuroendocrine state

Adult male hamsters ($n = 35$) were used to determine the impact of triiodothyronine on *Sst* expression. Hamsters were transferred from LP to SP for 12 weeks ($n = 24$) to initiate circannual interval timing and establish the maintenance phase. Hamsters were divided into two groups for the last week of SP treatment. A SP reference group received subcutaneous injections of 0.9% wt/vol saline for 7 days ($n = 10$). To determine the sufficiency of triiodothyronine to regulate *Sst* expression, hamsters ($n = 11$) were administered a single subcutaneous injection of 5 µg/100 µl triiodothyronine (Merck 102467157) prior to lights off on the final night. Subcutaneous injections of triiodothyronine are well established to regulate transcript expression in the hamster hypothalamus (*Banks et al., 2016*; *Bao et al., 2019*; *Stevenson et al., 2014*). The subsequent day, hamsters were sacrificed by cervical dislocation followed by exsanguination between 4 and 5 hr after lights on. Terminal body mass and epididymal adipose tissue mass were recorded using a Sartorius cp64 anatomical balance. Brains were dissected and stored immediately at –70°C until sectioning. Brains were cut into 200 µM sections using a Leica CM1520 cryostat. Anatomical structures (optic tract to the infundibular stem; Bregma –2.12 to –3.80mm) were used to isolate the MBH. Bilateral tissue punches were performed using an Integra Miltex 1 mm disposable biopsy punch. Tissue punches were stored at –70 °C until qPCR assays (see below).

## Experiment 4 – Functional manipulation of *Dio3* expression in hamsters

Targeted neuroanatomical injections, using highly precise Guide RNA (gRNA) against *Dio3* and Crispr–Cas9 vectors, were essential to ensure the long-term inhibition of *Dio3* function in the MBH in hamsters; an approach necessary to reliably examine the functional significance of *Dio3* for the duration of circannual interval timing. Custom-made Crispr–Cas9 constructs were packaged in a lentiviral vector by Merck Life Science UK. Lentiviral vectors are established to effectively transfect in adult Syrian hamsters (*Gao et al., 2014*) and in Siberian hamsters (*Munley et al., 2022*). The lentiviral vector used was U6-gRNA:ef1a-puro-2A-Cas9-2A-tGFP (Sigma All-in-One vector). gRNA for *Dio3* was designed using CHOPCHOP (*Labun et al., 2019*). Three gRNAs were designed to target the *Mus musculus Dio3* gene (gRNA1: CGACAACCGTCTGTGCACCCTGG; gRNA2: GTTCCCGCGCTTCCTA GGCACGG, and gRNA3: GACCCAGCCGTCGGATGGGTGGG). Only gRNA 1 and 2 were identified to align with 5′ end of the Siberian hamster *Dio3* gene and were predicted to align with chromosome 12 at locations 110279543 and 110279375, respectively (*Figure 3—figure supplement 1*). To check the specificity of gRNA sequences, we conducted BLAST using the NCBI dataset. Both gRNA1 and gRNA2 were found to align 100% with *Dio3* and did not align 23/23 with any other region.

Adult male hamsters ($n = 10$) were selected to assess the impact of *Dio3* genome modification on circannual interval timing. Hamsters were kept on LP conditions to maintain the summer phenotype. Hamsters were then divided into two treatment groups. The control group included hamsters ($n = 6$) that received ICV injections of blank Crispr–Cas9 constructs that were packaged into a lentivirus *Dio3*[wt]; U6-gRNA:ef1a-puro-2A-Cas9-2A-tGFP (Sigma All-in-One vector). The treatment group consisted of hamsters ($n = 4$) that received an ICV injection of Crispr–Cas9 constructs that harbored both gRNA1 and gRNA2 (*Dio3*[cc]).

ICV injections provide the ability to induce neuroanatomically localized manipulations in *Dio3* gene function in the adult brain. Crispr–Cas9 constructs were delivered via stereotaxic ICV injection into the third ventricle to target the MBH tanycytes. Under general anesthesia (5% induction, 2% maintenance of isoflurane mixed with oxygen), a SGE series II syringe was positioned along the midline at Bregma –1.5 mm, to a depth of –5.7 mm. The anatomical coordinates were refined based on estimates taken

from adult hamster brains (*Steward et al., 2003*) and tested using fresh hamster cadavers. Analgesia was administered subcutaneously (5 mg/kg Rimadyl and 0.1 mg/kg buprenorphine). A small ~1 cm incision was made by scalpel to expose the skull and a small hole was drilled using a Bone Micro Drill for Brain surgery (Harvard Apparatus). Lentivirus was administered using a Pump 11 Elite Programmable Syringe injection system (Harvard Apparatus) and 1 μl of viral vector (2–2.5 E + 13 vg/ml) was delivered over 10 min at a rate of 0.1 μl per minute. To allow viral diffusion and prevent backflow, the syringe was kept in place after injection for 1 min and then the syringe was raised by 1 mm and kept in place for another 1 min. Animals were then moved to heated home cages and given mashed food to aid recovery. Hamsters were provided with 2 weeks recovery and were monitored to ensure stable body mass, locomotor activity, and continued food and water intake.

To assess the impact of *Dio3$^{cc}$* on circannual interval timing, hamsters were moved to SP for 32 weeks and body mass and pelage were measured biweekly. Pelage score was determined using visual assessment and defined scale in which a score of 1 is the summer dark agouti coat; a score of 2 is a transition between agouti and white; and a score of 3 is a full winter white pelage (*Marshall et al., 2024*). Body mass was monitored to the nearest 0.1 g using an Ohaus portable scale (TA301). After 32 weeks, hamsters were killed 4–5 hr after lights on by cervical dislocation followed by exsanguination. Brains were dissected and stored immediately at –70°C. Brains were cut into 200 μM sections using a Leica CM1520 cryostat. Anatomical structures (optic tract to the infundibular stem; Bregma –2.12 to –3.80 mm) were used to isolate the MBH. Bilateral tissue punches were performed using an Integra Miltex 1 mm disposable biopsy punch. Tissue punches were stored at –70°C. DNA was extracted from dissected tissues using the DNeasy Blood & Tissue Kit (QIAGEN) as per the manufacturer's instructions. The *Dio3* gene was amplified using OneTaq Quick-Load 2X mastermix, as per the manufacturer's instructions (New England Biolabs). Amplification was achieved using a SimpliAmp thermal cycler (Applied Biosystems) the following thermal cycling conditions (94°C for 30 s (94°C for 15 s, 62°C for 30 s, 68°C for 3 min) for 45 cycles, 68°C for 5 min, 4°C until further analysis). The resultant amplicon was purified using the Qiaquick PCR purification kit (QIAGEN) as per the manufacturer's instructions. Isolated *Dio3* PCR amplicons were sequenced using the Eurofins LightRun Sanger sequencing service. The forward primer provided to Eurofins was CATGCTCCGCTCCCTGCTGCTTCA (5'–3'). CRISPR-modified transcripts were aligned to a reference wild-type transcript (animal #821) using the Tracy Sanger basecaller and aligner (*Rausch et al., 2020*). Alignments were visualized using the Ugene integrated bioinformatics tool (*Okonechnikov et al., 2012*). In case mutation was not obvious, an additional analysis was carried out using the associated online resource SAGE (https://www.gear-genomics.com/), to investigate the decomposition error of the modified transcripts with the wild-type control (821). Four *Dio3$^{wt}$* hamsters had *Dio3* sequences that fully matched the reference genome (*Figure 3—figure supplement 1*). Two hamster *Dio3* sequences did not automatically align, and the raw reads were visually inspected. No mutations or sequence mutations were detected in the raw sequence trace. Three *Dio3$^{cc}$*-treated hamsters had *Dio3* sequences with evidence of genomic mutation evidenced by low decomposition error (*Figure 3—figure supplement 1*). One *Dio3$^{cc}$*-treated hamster did not show any genomic mutation (905) and was removed from the final analysis. Hamster 905 had a body mass circannual interval period of 32 weeks and we propose the animal represents a false positive control.

## Experiment 5 – Naturally occurring variation in *Dio3* expression and photoperiodic response

A subset of hamsters does not physiologically respond to SP treatment and is classified as photoperiodically non-responsive (NR) (*Przybylska-Piech and Jefimow, 2022*). Adult hamsters ($n$ = 14) were held in LP or SP for 8 weeks. A group of male non-responders ($n$ = 4) was identified due to the absence of body mass and pelage change in response to SP (*Figure 3e*). A reference group of male SP responders ($n$ = 10) was collected. MBH was extracted, RNA isolated, and cDNA synthesized. qPCR was carried out for *Dio3* and *Sst* expression. Hamsters were killed between 2 and 4 hr after lights on.

## RNA extraction

RNA was extracted using the QIAGEN RNEasy plus mini kit. Tissues were homogenized using a Polytron PT 1200 E. RNA was then extracted from homogenized tissue following the manufacturer's instructions. RNA was tested for quality and quantity using an ND-1000 Nanodrop.

**Table 1.** PCR primer and qPCR parameters.

| Gene | Primer | Size | Temp (°C) | Melt (°C) |
|------|--------|------|-----------|-----------|
| Reference | | | | |
| *18s* | GCTCCTCTCCTACTTGGATAACTGTG | 111 | 62 | 80 |
| | CGGGTTGGTTTTGATCTGATAAATGCA | | | |
| *Hrpt* | AGTCCCAGCGTCGTGATTAGTGATG | 141 | 62 | 76 |
| | CGAGCAAGTCTTTCAGTCCTGTCCA | | | |
| Mediobasal hypothalamus | | | | |
| *Npy* | CCAGGCAGAGATACGGCAAGAGATC | 119 | 60 | 81 |
| | CCATCACCACATGGAAGGGTCC | | | |
| *Sst* | GAAGTCTCTGGCGGCTGCTG | 145 | 60 | 85 |
| | CAGCCTCATTTCATCCTGCTCCG | | | |
| *Dio3* | CATGCTCCGCTCCCTGCTGCTTCA | 251 | 62 | 85 |
| | CAGGGTGCACAGACGGTTGTC | | | |
| Pituitary gland | | | | |
| *Prl* | TCCGGAAGTCCTTCTGAACC | 300 | 60 | 82 |
| | CGCAGGCAGCGAATCTTATTG | | | |
| *Gh* | ACCTACAAAGAGTTTGAGCGTG | 167 | 58 | 85 |
| | ATGAGCAGCAGCGAGAATCG | | | |

## Next-generation sequencing and data processing

Oxford Nanopore Sequencing was used to carry out transcriptomic sequencing. RNA from MBH, DMH, and PVN samples was sequenced using the PCR-cDNA barcoding kit following the manufacturer's instructions (SQK-PCB109; Oxford Nanopore). RNA from the pituitary gland samples was synthesized using the direct cDNA sequencing kit following the manufacturer's instructions (SQK-DCS109 with EXP-NBD104; Oxford Nanopore). Bioinformatic analysis was carried out on Linux within a conda environment. Raw reads (Fast5) were demultiplexed and basecalled using guppy basecaller (4.2.1). Porechop (0.2.4) was used to remove adapters from reads and Filtlong (v0.2.0) was used to filter for quality and read length (length: >25 base pairs, quality score >9). Transcripts were aligned to *M. musculus* genome (GRCm39) using Minimap2 (*Li, 2018*). Previously, rodent genomes have been demonstrated to show significant similarity, such that cross-species transcriptomic analysis appears feasible (*Przybylska-Piech and Jefimow, 2022*). Transcript expression levels were generated using Salmon (v0.14.2) (*Patro et al., 2017*) and EdgeR (v3.24.3) (*Robinson et al., 2010*) was used to filter lowly expressed transcripts.

## cDNA synthesis and qPCR

RNA was transformed into cDNA for qPCR analysis using Superscript III (Invitrogen) as per the manufacturer's instructions. Quantification of transformed cDNA was achieved using Brilliant II SYBR Mastermix (Agilent). Stock forward and reverse primers (100 pmol/μl) were mixed and diluted in nuclease-free water to 20 pmol/μl. A working SYBR mixture was prepared by mixing 1-part primer mixture per 24-part SYBR mastermix. Reaction mixtures were prepared in wells on a 96-well plate by mixing 4.8 μl of normalized sample and 4.8 μl of SYBR working mix. All reactions were performed in duplicate. Primer and qPCR parameters are outlined in *Table 1*. qPCR reactions were carried out in a Stratagene Mx3000P thermal cycler. Cycling conditions utilized were in sequence; at 95°C for 5 min (denaturing), 40 cycles at (95°C for 30 s, X°C for 1 min – see *Figure 1—figure supplement 1*, 72°C for 30 s with fluorescent measurement at end), 95°C for 1 min, 55°C for 30 s increasing to 95°C (Melt curve analysis). Melt cure analysis was used to determine specificity of amplification. Data was

analyzed for meanCT, efficiency, and variability using PCR Miner (*Zhao and Fernald, 2005*). Logfold CT was calculated using the ΔΔCT method using 18 s and/or *Hrpt* reference transcripts as these transcripts are stable across photoperiodic conditions (*Stewart et al., 2022*).

## Statistical analysis

Raw data is provided in *Source data 5* and bioinformatic analyses are provided in *Source data 2–4*. Physiological and qPCR datasets were tested for normality using the Shapiro test and transformed by log transformation if normality was violated. Statistical significance was determined using a two-way ANOVA. Post hoc analysis of significant data was achieved by *t*-test. Type 1 error was minimized by utilizing the Bonferroni adjustment on post hoc analyses. Data were analyzed using non-linear regression for rhythmicity using the online resource BioDare 2.0 (*Zielinski et al., 2014*) (biodare2.ed.ac.uk). The empirical JTK_CYCLE method was used for detection of rhythmicity and the classic BD2 waveform set was used for comparison testing. Rhythmicity was determined by a Benjamini–Hochberg controlled false discovery rate (BH corrected FDR <0.1). The false discovery rate of this dataset was controlled by applying the Benjamini–Hochberg method (*Benjamini and Hochberg, 1995*). Clusters within the dataset were identified using the gap statistic and clustered using *k*-means clustering. Gene ontology analysis was carried out using ShinyGO v0.77 (*Ge et al., 2020*).

## Acknowledgements

The authors thank Gerald Lincoln for helpful discussions which contributed to the conceptualization of the research. The authors thank the technical assistance provided by Nicola Munro, Ana Monterio, and Fallon Cuthill. Funding was provided via Leverhulme Trust award LT-RL-2019-06 (TJS) and ISSF Wellcome Trust award (TJS, PJM, and FJPE).

## Additional information

### Funding

| Funder | Grant reference number | Author |
| --- | --- | --- |
| Leverhulme Trust | LT-RL-2019-06 | Tyler J Stevenson |
| Wellcome Trust | Institutional Strategic Support Fund | Peter J Morgan<br>Fran JP Ebling<br>Tyler J Stevenson |

The funders had no role in study design, data collection, and interpretation, or the decision to submit the work for publication. For the purpose of Open Access, the authors have applied a CC BY public copyright license to any Author Accepted Manuscript version arising from this submission.

### Author contributions

Calum Stewart, Formal analysis, Validation, Investigation, Visualization, Methodology; T Adam Liddle, Data curation, Formal analysis, Investigation; Elisabetta Tolla, Data curation, Investigation; Jo Edward Lewis, Resources, Data curation, Investigation; Christopher Marshall, Investigation; Neil P Evans, Resources, Supervision, Writing – review and editing; Peter J Morgan, Supervision, Funding acquisition, Writing – review and editing; Fran JP Ebling, Resources, Supervision, Funding acquisition, Writing – review and editing; Tyler J Stevenson, Conceptualization, Formal analysis, Supervision, Funding acquisition, Visualization, Writing – original draft, Project administration, Writing – review and editing

### Author ORCIDs

Calum Stewart ⓘ https://orcid.org/0000-0002-6216-6044
Elisabetta Tolla ⓘ https://orcid.org/0009-0002-3425-233X
Jo Edward Lewis ⓘ https://orcid.org/0000-0002-5722-6778
Christopher Marshall ⓘ https://orcid.org/0000-0002-5658-9817
Peter J Morgan ⓘ https://orcid.org/0000-0002-5071-6512
Tyler J Stevenson ⓘ https://orcid.org/0000-0003-2644-9685

## Ethics

All procedures were in accordance with the National Centre for the Replacement, Refinement and Reduction of Animals in Research ARRIVE guidelines (https://www.nc3rs.org.uk/revision-arrive-guidelines). All procedures were approved by the Animal Welfare and Ethics Review Board at the University of Glasgow and conducted under the Home Office Project License PP5701950.

Reviewer #1 (Public review): https://doi.org/10.7554/eLife.106383.3.sa1
Reviewer #2 (Public review): https://doi.org/10.7554/eLife.106383.3.sa2
Author response https://doi.org/10.7554/eLife.106383.3.sa3

## Additional files

### Supplementary files

MDAR checklist

Source data 1. EdgR and BioDare2.0 analyses of Mediobasal Hypothalamus.

Source data 2. EdgR and BioDare2.0 analyses of the pituitary gland.

Source data 3. EdgR and BioDare2.0 analyses of the paraventricular nucleus.

Source data 4. EdgR and BioDare2.0 analyses of the dorsomedial hypothalamus.

Source data 5. Raw data for *Figures 1–3*.

### Data availability

All data are available in *Source data 1–5*. The code is available via GitHub (copy archived at *Stewart, 2026*). Sequencing information was deposited in GEO (GSE274003).

The following dataset was generated:

| Author(s) | Year | Dataset title | Dataset URL | Database and Identifier |
|---|---|---|---|---|
| Stewart C, Stevenson TJ | 2025 | Deiodinase type-3 establishes the period of circannual interval timing in mammals | https://www.ncbi.nlm.nih.gov/geo/query/acc.cgi?acc=GSE274003 | NCBI Gene Expression Omnibus, GSE274003 |

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
