## [Editor Report · eLife Assessment]

This study provides **important** findings on the understanding of circannual timing in mammals, for which iodothyronine deiodinases (DIOs) have been suggested to be of critical importance, yet functional genetic evidence has been missing. The authors **convincingly** implicate dio3, the major inactivator of the biologically active thyroid hormone T3, in circannual timing in Djungarian hamsters, using a combination of correlative and gene knock-out experiments; thus this provides key insights into the evolution and function of animal annual timing mechanisms.

---

## [Referee Report · Reviewer #1 (Public review)]

Circannual timing is a phylogenetically widespread phenomenon in long-lived organisms and is central to the seasonal regulation of reproduction, hibernation, migration, fur color changes, body weight, and fat deposition in response to photoperiodic changes. Photoperiodic control of thyroid hormone T3 levels in the hypothalamus dictates this timing. However, the mechanisms that regulate these changes are not fully understood. The study by Stewart et al. reports that hypothalamic iodothyronine deiodinase 3 (Dio3), the major inactivator of the biologically active thyroid hormone T3, plays a critical role in circannual timing in the Djungarian hamster. Overall, the study yields important results for the field and is well-conducted, with the exception of the CRISPR/Cas9 manipulation.

Comments on revisions:

The authors have satisfactorily addressed all my comments. I no longer have concerns about the CRISPR/Cas9 experiments which have been conducted properly and are now reported appropriately.

---

## [Referee Report · Reviewer #2 (Public review)]

Summary:

Several animals and plants adjust their physiology and behavior to seasons. These changes are timed to precede the seasonal transitions, maximizing chances of survival and reproduction. The molecular mechanisms used for this process are still unclear. Studies in mammals and birds have shown that the expression of deiodinase type-1, 2, and 3 (Dio1, 2, 3) in the hypothalamus spikes right before the transition to winter phenotypes. Yet, whether this change is required or an unrelated product of the seasonal changes has not been shown, particularly because of the genetic intractability of the animal models used to study seasonality. Here, the authors show for the first time a direct link between Dio3 expression and the modulation of circannual rhythms.

The work is concise and presents the data in a clear manner. The data is, for the most part, solid and supports the author's main claims. The use of CRISPR is a clear advancement in the field. This is, to my knowledge, the first study showing a clear (i.e., causal) role of Dio3 in the circannual rhythms in mammals. Having established a clear component of the circannual timing and a clean approach to address causality, this study could serve as a blueprint to decipher other components of the timing mechanism. It could also help to enlighten the elusive nature of the upstream regulators, in particular, on how the integration of day length takes place, maybe within the components in the Pars tuberalis, and the regulation of tanycytes.

Comments on revisions:

The authors have provided an improved version of the manuscript, particularly clarifying the methodology for their CRISPR manipulations. I am satisfied with their response and commend the authors for their work.

---

## [Author Response]

The following is the authors’ response to the previous reviews.

**Reviewer #1 (Public review):**
Circannual timing is a phylogenetically widespread phenomenon in long-lived organisms and is central to the seasonal regulation of reproduction, hibernation, migration, fur color changes, body weight, and fat deposition in response to photoperiodic changes. Photoperiodic control of thyroid hormone T3 levels in the hypothalamus dictates this timing. However, the mechanisms that regulate these changes are not fully understood. The study by Stewart et al. reports that hypothalamic iodothyronine deiodinase 3 (Dio3), the major inactivator of the biologically active thyroid hormone T3, plays a critical role in circannual timing in the Djungarian hamster. Overall, the study yields important results for the field and is well-conducted, with the exception of the CRISPR/Cas9 manipulation.

We appreciate the positive and supportive comment from the Reviewer. We have clarified the oversight in the Crispr/Cas9 data representation below. Our correction should alleviate any concern raised.

Figure 1 lays the foundation for examining circannual timing by establishing the timing of induction, maintenance, and recovery phases of the circannual timer upon exposure of hamsters to short photoperiod (SP) by monitoring morphological and physiological markers. Measures of pelage color, torpor, body mass, plasma glucose, etc, established that the initiation phase occurred by weeks 4-8 in SP, the maintenance by weeks 12-20, and the recovery after week 20, where all morphological and physiological changes started to reverse back to long photoperiod phenotypes.The statistical analyses look fine, and the results are unambiguous.

We thank the Reviewer for recognizing our attempts to highlight the phenomenon of circannual interval timing.

Their representation could, however, be improved. In Figures 1d and 1e, two different measures are plotted on each graph and differentiated by dots and upward or downward arrowheads. The plots are so small, though, that distinguishing between the direction of the arrows is difficult. Some color coding would make it more reader-friendly. The same comment applies to Figure S4.

We have increased the panel size for Figure 1d and 1e. We have also changed the colour of the graphs in Figure 1d and 1e to facilitate the differentiation of the two dependent variables. For the circos plots, we attempted different ways to represent the data. We have opted to keep the figures in their current stage. The overall aim is to provide a ‘gestalt’ view of the timing of changes in transcript expression and highlighted only a few key genes. The whole dataset is provided in the supplementary materials for Reviewer/Reader interrogation.

The authors went on to profile the transcriptome of the mediobasal and dorsomedial hypothalamus, paraventricular nucleus, and pituitary gland (all known to be involved in seasonal timing) every 4 weeks over the different phases of the circannual interval timer. A number of transcripts displaying seasonal rhythms in expression levels in each of the investigated structures were identified, including transcripts whose expression peaks during each phase. This included two genes of particular interest due to their known modulation of expression in response to photoperiod, Dio3 and Sst, found among the transcripts upregulated during the induction and maintenance phases, respectively. The experiments are technically sound and properly analyzed, revealing interesting candidates. Again, my main issues lie with the representation in the figure. In particular, the authors should clarify what the heatmaps on the right of Figures 1f and 1g represent. I suspect they are simply heatmaps of averaged expression of all genes within a defined category, but a description is missing in the legend, as well as a scale for color coding near the figure.

We have clarified the heatmap and density maps in the Figure legend. We apologise for the lack of information to describe the figure panels. (see lines 644-648)

Figure 2 reveals that SP-programmed body mass loss is correlated to increased Dio3-dependent somatostatin (Sst) expression. First, to distinguish whether the body mass loss was controlled by rheostatic mechanisms and not just acute homeostatic changes in energy balance, experiments from hamsters fed ad lib or experiencing an acute food restriction in both LP and SP were tested. Unlike plasma insulin, food restriction had no additional effect on SP-driven epididymal fat mass loss (Figure S7). This clearly establishes a rheostatic control of body mass loss across weeks in SP conditions. Importantly, Sst expression in the mediobasal hypothalamus increased in both ad lib fed or restriction fed SP hamsters and this increase in expression could be reduced by a single subcutaneous injection of active T3, clearly suggesting that increase in Sst expression in SP is due to a decrease of active T3 likely via Dio3 increase in expression in the hypothalamus. The results are unambiguous

We thank the Reviewer for the supportive and affirmative feedback.

Figure 3 provides a functional test of Dio3's role in the circannual timer. Mediobasal hypothalamic injections of CRISPR-Cas9 lentiviral vectors expressing two guide RNAs targeting the hamster Dio3 led to a significant reduction in the interval between induction and recovery phases seen in SP as measured by body mass, and diminished the extent of pelage color change by weeks 15-20. In addition, hamsters that failed to respond to SP exposure by decreasing their body mass also had undetectable Dio3 expression in the mediobasal hypothalamus. Together, these data provide strong evidence that Dio3 functions in the circannual timer. I noted, however, a few problems in the way the CRISPR modification of Dio3 in the mediobasal hypothalamus was reported in Figure S8. One is in Figure S8b, where the PAM sites are reported to be 9bp and 11bp downstream of sgRNA1 and sgRNA2, respectively. Is this really the case? If so, I would have expected the experiment to fail to show any effect as PAM sites need to immediately follow the target genomic sequence recognized by the sgRNA for Cas9 to induce a DNA double-stranded break. It seems that each guide contains a 3' NGG sequence that is currently underlined as part of sgRNAs in both Fig S8b and in the method section. If this is not a mistake in reporting the experimental design, I believe that the design is less than optimal and the efficiencies of sgRNAs are rather low, if at all functional.

We apologize for the oversight and indeed the reporting in Figure S8b was a mistake. The PAM site previously indicated was the ‘secondary PAM site’ (which as the Reviewer notes would likely have low efficiency). The PAM site is described within the gRNA in the figure. We use Adobe Illustrator to generate figures, and during the editing process, the layer for PAM text was accidentally moved ‘back’ to a lower level. The oversight was not rectified before submission. We apologise for this unreservedly. The PAM site text has been moved forward, to highlight the location of the primary site (ie immediately following gRNA) and labelled the gRNA and PAM site in the ‘Target region’. The secondary PAM site text was removed to eliminate any confusion.

The authors report efficiencies around 60% (line 325), but how these were obtained is not specified.

The efficiency provided are based on bioinformatic analyses and not in vivo assays. To reduce any confusion, we have removed the text. The gRNA were clearly effective to induce mutations based on the sequencing analyses.

Another unclear point is the degree to which the mediobasal hypothalamus was actually mutated. Only one mutated (truncated) sequence in Figure S8c is reported, but I would have expected a range of mutations in different cells of the tissue of interest.

The tissue punch would include multiple different cells (e.g., neuronal, glial, etc). We agree with the Reviewer that genomic samples from different cells would be included in the sequencing analyses. Given the large mutation in the target region, the gRNA was effective. We have only shown one representative sequence. If the Reviewer would like to see all mutations, we can easily show the other samples.

Although the authors clearly find a phenotypic effect with their CRISPR manipulation, I suspect that they may have uncovered greater effects with better sgRNA design. These points need some clarification. I would also argue that repeating this experiment with properly designed sgRNAs would provide much stronger support for causally linking Dio3 in circannual timing.

The gRNA was designed using the Gold-standard approach – ChopChop [citation Labon et al., 2019]. If the Reviewer’s concern re design is due to the comment above re PAM site; this issue was clarified and there are no concerns for the gRNA design. The major challenge with the Dio3 gene (single exon) with a very short sequence length (approx.. 412bp). There is limited scope within this sequence length to generate gRNA.

A proposed schematic model for mechanisms of circannual interval timing is presented in Figure S9. I think this represents a nice summary of the findings put in a broader context and should be presented as a main figure in the manuscript itself rather than being relayed in supplementary materials.

We agree with the Reviewer position and moved the figure to the main manuscript. The figure is now Figure 4.

**Reviewer #2 (Public review):**
Several animals and plants adjust their physiology and behavior to seasons. These changes are timed to precede the seasonal transitions, maximizing chances of survival and reproduction. The molecular mechanisms used for this process are still unclear. Studies in mammals and birds have shown that the expression of deiodinase type-1, 2, and 3 (Dio1, 2, 3) in the hypothalamus spikes right before the transition to winter phenotypes. Yet, whether this change is required or an unrelated product of the seasonal changes has not been shown, particularly because of the genetic intractability of the animal models used to study seasonality. Here, the authors show for the first time a direct link between Dio3 expression and the modulation of circannual rhythms.

We appreciate the clear synthesis and support for the manuscript.

Strengths:The work is concise and presents the data in a clear manner. The data is, for the most part, solid and supports the author's main claims. The use of CRISPR is a clear advancement in the field. This is, to my knowledge, the first study showing a clear (i.e., causal) role of Dio3 in the circannual rhythms in mammals. Having established a clear component of the circannual timing and a clean approach to address causality, this study could serve as a blueprint to decipher other components of the timing mechanism. It could also help to enlighten the elusive nature of the upstream regulators, in particular, on how the integration of day length takes place, maybe within the components in the Pars tuberalis, and the regulation of tanycytes.

We thank the Reviewer for this positive summary.

Weaknesses:Due to the nature of the CRISPR manipulation, the low N number is a clear weakness. This is compensated by the fact that the phenotypes shown here are strong enough. Also, this is the only causal evidence of Dio3's role; thus, additional evidence would have significantly strengthened the author's claims. The use of the non-responsive population of hamsters also helps, but it falls within the realm of correlations.

We would also like to remind the Reviewer that one Crispr-Cas9 Dio3^cc^ treated hamster did not show any mutation in the genome. This hamster was observed to have a change in body mass and pelage colour like controls. This animal provides another positive control.

We also conducted a statistical power analysis to examine whether n=3 is sufficient for the Dio3^cc^ treatment group. Using the appropriate expected difference in means and standard deviations for an alpha of 0.05; we regularly observed beta >0.8 across the dependent variables.

Additionally, the consequences of the mutations generated by CRISPR are not detailed; it is not clear if the mutations affect the expression of Dio3 or generate a truncation or deletion, resulting in a shorter protein.

We agree with the Reviewer that transcript and protein assays would strengthen the genome mutation data. Due to the small brain region under investigation, we are limited in the amount of biological material to extract. Dio3 is an intronless gene and very short – approximately 412 base pairs in length. We opted to maximize resources into sequencing the gene as the confirmation of genetic mutation is paramount. Given the large size of the mutation in the treated hamsters, there would be no amplification of transcript or protein translated.

**Reviewer #3 (Public review):**
The authors investigated SP-induced physiological and molecular changes in Djungarian hamsters and the endogenous recovery from it after circa half a year. The study aimed to elucidate the intrinsic mechanism and included nice experiments to distinguish between rheostatic effects on energy state and homeostatic cues driven by an interval timer. It also aimed to elucidate the role of Dio3 by introducing a targeted mutation in the MBH by ICV. The experiments and analyses are sound, and the amount of work is impressive. The impact of this study on the field of seasonal chronobiology is probably high.

We thank the Reviewer for their positive comments and support for our work.

Even though the general conclusions are well-founded, I have fundamental criticism concerning 3 points, which I recommend revising:(1) The authors talk about a circannual interval timer, but this is no circannual timer. This is a circasemiannual timer. It is important that the authors use precise wording throughout the manuscript.

We agree with the Reviewer that the change in physiology and behaviour does not approximate a full year (e.g. annual) and only a half of the year. We opted to use circannual timer as this term is established in the field (see doi: 10.1177/0748730404266626; doi: 10.1098/rstb.2007.2143). We cannot identify any publication that has used the term ‘semiannual timer’. We do not feel this manuscript is the appropriate time to introduce a new term to the field; we will endeavour to push the field to consider the use of ‘semiannual timer’. A Review or Opinion paper is best place for this discussion. We hope the Reviewer will understand our position.

(2) The authors put their results in the context of clocks. For example, line 180/181 seasonal clock. But they have described and investigated an interval timer. A clock must be able to complete a full cycle endogenously (and ideally repeatedly) and not only half of it. In contrast, a timer steers a duration. Thus, it is well possible that a circannual clock mechanism and this circa-semiannual timer of photoperiodic species are 2 completely different mechanisms. The argumentation should be changed accordingly.

We agree with the Reviewers definitions of circannual ‘clock’ and ‘timer’. We were careful to distinguish between the two concepts early in the manuscript (lines 41-46). We have added italics to emphasis the different terms. The use of seasonal clock on line 180/191 was imprecise and we appreciate the Reviewer highlighting our oversight and the text was revised. We have also revised the Abstract accordingly.

(3) The authors chose as animal model the Djungarian hamster, which is a predominantly photoperiodic species and not a circannual species. A photoperiodic species has no circannual clock. That is another reason why it is difficult to draw conclusions from the experiment for circannual clocks. However, the Djungarian hamster is kind of "indifferent" concerning its seasonal timing, since a small fraction of them are indeed able to cycle (Anchordoquy HC, Lynch GR (2000), Evidence of an annual rhythm in a small proportion of Siberian hamsters exposed to chronic short days. J Biol Rhythms 15:122-125.). Nevertheless, the proportion is too small to suggest that the findings in the current study might reflect part of the circannual timing. Therefore, the authors should make a clear distinction between timers and clocks, as well as between circa-annual and circa-semiannual durations/periods.

This comment is not clear to us. The Reviewer states the hamsters are not a circannual species, but then highlight one study that shows circannual rhythmicity. We agree that circannual rhythmicity in Djungarian hamsters is dependent on the physiological process under investigation (e.g. body mass versus reproduction) and that photoperiodic response system either dampen or mask robust cycles. We have corrected the text oversight highlighted above and the manuscript is focused on interval timers. We have kept the term circannual over semicircannual due to the prior use in the scientific literature.

**Reviewing Editor Comments:**
The detailed suggestions of the reviewers are outlined below (or above in case of reviewer 1). In light of the criticism, we ask the authors to especially pay attention to the comments on the Cas9/Crisp experiment, raised by Reviewers 1 and 2. As currently described, there are serious questions on the design of the sgRNAs, and also missing critical methodological details. If the latter are diligently taken care of, they may resolve the questions on the sgRNA design. Please also reconsider the wording along the suggestions of Reviewer 3.

We appreciate the Editors time and support for the manuscript. We have clarified and corrected our oversight for the PAM site. This correction confirms the strength of the Crispr-cas9 gRNA used in the study. The correction should remove all concerns. We have also considered using semicircannual in the text. As there is existing scientific literature using circannual interval timer, and there is no publication to our knowledge for using ‘semicircannual; we have opted to keep with the current approach and use circannual. We feel a subsequent Opinion paper is more suitable to introduce a new term.

**Reviewer #2 (Recommendations for the authors):**
First, I want to commend the authors for their work. It is a clear advancement for our field. Below are a couple of comments and suggestions I have:

we thank the Review for the positive comment and support. We have endeavoured to incorporate their suggested improvements to the manuscript.

(1) Looking at the results of Figure 1A and Figure S8, the control in S8 showed a lower pelage color score as compared to the hamsters in 1A. Is this a byproduct of the ICV injection?

The difference between Figure 1 and 3 is likely due to the smaller sample sizes. The controls in Figure 1 had a higher proportion of hamsters show complete white fur (score = 3) at 1618 weeks compared to controls in Figure 3. It is possible, although unlikely that the ICV injection would reduce the development of winter phenotype. There was no substance in the ICV injection that would impact the prolactin signalling pathway. Our perspective is that the difference between the two figures is due to the different sampling population. Overall, the timing of the change in pelage colour is the same between the figures and suggest that the mechanisms of interval timer were unaffected.

(2) Is there a particular reason why the pelage color for the CRISPR mutants is relegated to the supplemental information? In my opinion, this is also important, even though the results might be difficult to explain. Additionally, did the authors check for food intake and adipose mass in these animals?

We agree with the Reviewer the pelage change is very interesting. We decided to have Figure 3 focus on body mass. The rationale was due to the robust nature of the data collection from Crispr-cas9 study (Fig.3b), in addition to the non-responsive hamsters (Fig.3e). We disagree that the data patterns are hard to explain, as pelage changes was similar to the photoperiodic induced change in body mass. No differences were observed for food intake or adipose tissue. We have added this information in the text (see lines 162-163).

(3) I might have missed it, but did the authors check for the expression of Dio3 on the CRISPR mutants? Does the deletion cause reduced expression or any other mRNA effect, such as those resulting in the truncation of a protein?

Due to the limited biological material extracted from the anatomical punches, we decided to focus on genomic mutations. Dio3 has a very short sequence length and the size of the mutations identified indicate that no RNA could be transcribed.

(4) Could the authors clarify which reference genome or partial CDS (i.e., accession numbers) they used to align the gRNA? Did they use the SSSS strain or the Psun_Stras_1 isolate?

The gRNAs were designed using the online tool CHOPCHOP, using the *Mus musculus*

Dio3 gene. The generated gRNAs were subsequently aligned via blast with the Phodopus sungorus Dio3 partial cds (GenBank: MF662622.1), to ensure alignment with the species. We are confident that the gRNA designed align 100% in hamsters. Furthermore, we conducted BLAST to ensure there were no off-targets. The only gene identified in the BLAST was the rodent (i.e. hamster, mouse) Dio3 sequence.

(5) Figure 3b. I do agree with the authors in pointing out that the decrease in body mass is occurring earlier in Dio3wt hamsters; however, the shape of the body mass dynamic is also different. Do the authors have any comments on the possible role of Dio3 in the process of exist of overwintering?

This is a very interesting question. We do not have the data to evaluate the role of Dio3 for overwintering. We argue that disruption in Dio3 reduced the circannual interval period. For this interpretation, yes, Dio3 is necessary for overwintering. However, we would need to show the sufficiency of Dio3 to induce the winter phenotype in hamsters housed in long photoperiod. At this time, we do not have the technical ability to conduct this experiment.

(6) In Figure 3d, the Dio3wt group does not show any dispersion. Is this correct? If that's true, and no dispersion is observed, no normality can be assumed, and a t-test can't be performed (Line 692).The Mann-Whitney test might be better suited.

We conducted a Welch’s t-test to compare the difference in body mass period. We used the Welch’s test as the variance were not equal; Mann-Whitney test is best for skewed distributions. To clarify the test used, we have added ‘Welch’s test’ to the Figure legend.

(9) Figure 1 h. It might be convenient to add the words "Induction", "maintenance", and "recovery" over each respective line on the polar graph for easier reading.

We have added the text as suggested by the Reviewer.

**Reviewer #3 (Recommendations for the authors):**
(1) Figure 1: Please enlarge all partial graphics at least to the size of Figure 2. In the print version, labels are barely readable

we have increased the panels in Figure 1 and 3 by 20% to accommodate the Reviewers suggestion.

(2) Legend Figure 2: Add that the food restriction was 16h.

We have added 16h to the text.

(3) Figure 3b: enlarge font size. In the legend: Dio3cc hamsters delayed.... The delay might have been a week or so, but not more (and even that is unclear since the rise in body mass in that week seems to be rather a disturbance of the curve). Thus 'delay' might not be the most appropriate wording. Instead, the initial decline is slower, but both started at nearly the same week (=> no delay). Minimum body mass is reached at the identical week as in wt (=> no delay). Also, the increase started at the same week but was much faster in Dio3cc than in wt. Figure 3c: How can there be a period when there is no repeated cycle (rhythm)? This is rather a duration. Moreover, according to the displayed data, I am wondering which start point and which endpoint is used. The first and last values are the highest of the graph, but have they been the maximum? Especially for Dio3wt, it can be assumed that animals haven't reached the maximum at the end of the graph.

We have increased the font size in Figure 3b. We have changed ‘delayed’ to ‘slower’ in the text. Period analyses, such as the Lomb-Scargle measure the duration of a cycle (and multiple cycles). The start point and end point used in the analyses were the initial data collection date (week 0) and the final data collection date (week 32). The Lomb-Scargle analyses determines the duration of the period that occurs within these phases of the cycle. We believe the period analyses conducted by the Lomb-Scargle is the most suitable for the scientific question.

(4) Figure S9: This is a very nice graph and summarises your main results. It should appear in the main manuscript and not in the supplements.

We appreciate the positive comment and suggestion. We agree with the Reviewer and have move the graph to the main figure. The revised manuscript indicates the graph as Figure 4.